

# Mobile phone addiction and non-suicidal self-injury among adolescents in China

Rui Wang[1,2,*], Runxu Yang[1,2,*], Hailiang Ran[3], Xiufeng Xu[1,2], Guangya Yang[4], TianLan Wang[4], Yusan Che[4], Die Fang[4], Jin Lu[1,2] and Yuanyuan Xiao[3]

[1] Psychiatric Department, The First Affiliated Hospital of Kunming Medical University, Kunming, Yunnan, China
[2] Yunnan Clinical Research Center for Mental Health, Kunming, Yunnan, China
[3] Department of Epidemiology and Health Statistics, School of Public Health, Kunming Medical University, Kunming, Yunnan, China
[4] Psychiatric Department, Lincang Psychiatric Hospita, Lincang, Yunnan, China
* These authors contributed equally to this work.

## ABSTRACT

**Background:** Non-suicidal self-injury (NSSI) has recently widely discussed. Independently, mobile phone addiction (MPA) has also attracted academic attention. A few research have examined the correlation between the two. However, there is inadequate knowledge to characterize this relationship altogether. This study further explores the correlation between MPA and NSSI, specifically repeated and severe NSSI.

**Method:** A population-based cross-sectional survey was conducted among 2,719 adolescents in Lincang, Yunnan. The mobile phone addiction index (MPAI) and the Modified Adolescents Self-Harm Survey (MASHS) were administered in combination. The connection between the MPAI and NSSI, as well as both repeated and severe NSSI, was studied using univariate and multivariate logistic regression models. (The copyright holders have permitted the authors to use the MPAI and the MASHS).

**Results:** The prevalence of NSSI was 47.11% (95% CI [36.2–58.0%]), and the detection rate of MPA was 11.11% (95% CI [6.7–18.0%]). The prevalence of NSSI among those with MPA was 4.280 times (95% CI [3.480–5.266]) that of respondents not exhibiting MPA. In addition, all subscales of the MPAI, except for the feeling anxious and lost subscale (FALS), were positively correlated with NSSI. Risk factors, represented by odds ratios, of repeated NSSI with the inability to control cravings subscale (ICCS), the FALS, and the withdrawal and escape subscale (WES) was 1.052 (95% CI [1.032–1.072]), 1.028 (95% CI [1.006–1.051]), and 1.048 (95% CI [1.019–1.078]) respectively. Risk factors of these same three subscales for severe NSSI, had odds ratios of 1.048 (95% CI [1.029–1.068]), 1.033 (95% CI [1.009–1.057]), and 1.045 (95% CI [1.018–1.073]).

**Conclusion:** MPA was shown to be a risk factor for NSSI in adolescents. Individuals with high scores on the ICCS, the WES, and the FALS were more prone to experience repeated and severe NSSI. As a result, early assessment using the MPAI to determine the need for intervention can contribute to the prediction and prevention of NSSI.

Corresponding authors
Jin Lu, Jinlu2000@163.com
Yuanyuan Xiao, 33225647@qq.com

# INTRODUCTION

Non-suicidal self-injury (NSSI) is defined as deliberate repetitive self-harm including that by cutting, burning, or scratching the skin, hitting or banging the wall, or by drug overdose without the intention of committing suicide. This behavior is typically not accepted by society (*Lang & Yao, 2018*; *Muehlenkamp et al., 2012*). Research to date has shown that adolescents have the highest incidence of NSSI, with an international prevalence of known cases ranging from 7% to 37.2%, and from 10.1% to 22.37% domestically (*Kwok et al., 2014*; *Lang & Yao, 2018*; *Zhang et al., 2016*). A survey of the international prevalence of adolescents NSSI reported a 12-month prevalence estimate of 7.3% for NSSI in the U.S. ($n = 61,767$) conducted by *Muehlenkamp et al. (2012)*. Asian countries have a greater overall lifetime prevalence of NSSI (32.6%) than Western countries (19.4%) (*Lim et al., 2019*). This studies among community samples have found that most people have at least one non suicidal self-injury behavior in their life, but the incidence varies among regions, which may potentially reveal different protection and risk factors for self-injury in different regions. The fast pace of society, the rapid development of the internet and social media, increasing amounts of stress, as well as the mainstream visibility of NSSI have all contributed to an upward trend in its incidence (*Morgan, Webb & Carr, 2017*). Current studies on NSSI have demonstrated associations with depression, anxiety, low self-esteem, and impulsivity, along with factors in the domestic and social environment (*Liu et al., 2018*). NSSI is also seen as a significant risk factor for suicide attempts and suicide deaths (*Hawton et al., 2020*). When compared to the clinical sample research results of adolescent patients who choose to come to a psychiatric department or seek emergency treatment because of self-injury, it is discovered that the incidence of NSSI is much higher than that of the general population, and that there is a higher risk of suicide (*Plener et al., 2015*). This may be related to the combination of specific psychiatric disorders. In the USA, adolescents hospitalized due to NSSI are 37.2 times more likely to commit suicide in the following year than their healthy peers (*Olfson et al., 2018*), suggesting that adolescents exhibiting repeated and severe self-injury are at higher risk of suicide. Therefore, NSSI attracts much attention as a sign of serious mental health problems, and it is imperative to develop a more comprehensive understanding that can be used in suicide prevention.

Mobile phones have become a requirement in most facets of life due to the rapid expansion of information technology and the continual innovation of mobile phone applications. While mobile phones have made many people's lives more convenient, provided entertainment and stress relief, and provided many students with the tools to maintain a connection with classmates and expand their social circles, mobile phone addition (MPA) has become a health concern (*Coskun & Karayagiz, 2019*). Research on MPA was first established by YouGov PLC Research Institute in the UK in 2001. Different sources use the synonyms "mobile phone dependence" "mobile phone addiction" and "problematic mobile phone use". At present, MPA has not been included in the DSM, so there is no unified definition, however, its central manifestation is the excessive use of a

mobile phone, such as the use of the phone constantly to attain a sense of fulfillment, which disrupts personal and social activities and causes mood swings if access to the phone is prohibited (*Huang et al., 2022*). The prevalence of MPA among Chinese college students is 21.3%, and that among British teenagers is 10% (*Long et al., 2016*; *Lopez-Fernandez et al., 2014*).

People who are addicted to their cellphones not only experience physical symptoms like headaches, tinnitus, fatigue, and insomnia, but also withdrawal-like symptoms like frustration, feeling lost, and lonely, which can lead to psychological and behavioral issues like anxiety, depression, aggression, and suicide, according to studies (*Augner & Hacker, 2012*; *Yang et al., 2010*). Numerous research identified quality of life, social self-efficacy, and self-esteem as predictors of MPA. Adverse childhood experiences (ACES) and addiction have also been associated in several studies (*Kumcagiz, 2018*; *Kim & Koh, 2018*; *Levenson, 2016*).

NSSI and MPA have been associated with factors affecting the mental health of adolescents, such as emotions, personality changes, and their environment. Several studies have investigated the nature of this association. One study demonstrated an independent correlation between NSSI and MPA. They found that many students reported using their phones in the evening and after lights out. Phone use during this time has been shown to affect sleep patterns and quality of sleep, which can, in turn, cause psychological and psychiatric issues. Interestingly, NSSI was still associated with MPA even after controlling for sleep-related confounds (*Li et al., 2019*; *Oshima et al., 2012*). Most existing studies investigating NSSI and mobile phone addition have done so in a population of college students, meaning that there is a lack of research in adolescents. The MPA index (MPAI) in China is extensively utilized in similar investigations. The full scale can be divided into four subscales: the inability to control cravings subscale (ICCS), the feeling anxious and lost subscale (FALS), the withdrawal and escape subscale (WES), and the productivity loss subscale (PLS).

This study uses a cross-sectional design to address the lack of research on the relationship between NSSI and MPA using the MPAI with a cross-sectional study design in a sample of students from Lincang. We make the following hypotheses: MPA and the four subscales of MPAI will have a positive linear association with NSSI and its rate of recurrence and severity. If these hypotheses are proven positive, a proper and tailored preventive guideline is needed to reduce the risk of NSSI and subsequent suicide in adolescents.

## MATERIALS AND METHODS

### Study setting

The study was approved by the Ethics Committee of Lincang Psychiatric Hospital (The Third People's Hospital of Lincang, Approval number: 2019-01) and conducted in accordance with the principles of the Declaration of Helsinki. All participants, and their legal guardians, gave written informed consent prior to participating in the study.

A cross-sectional survey was conducted in Lincang, a southwestern prefecture in Yunnan, China, between December 1 and 13, 2019. A three-stage random cluster sampling

was used. In the first stage, Linxiang district was randomly selected from eight districts and counties in Lincang. In stage two, five primary schools, five middle schools, and four high schools were randomly selected. In the final stage, either three or four classes were randomly selected from each school, based on the required sample size. All eligible students living in the survey area for at least 6 months per year were preliminarily included in the sample pool, but any student who was not aged between 10 and 18, or with a hearing or communication impairment was excluded from the study. It is important to emphasize that children too young may not accurately understand the meaning of self-harm and suicide (*Mishara, 1999*), so we set the lower age limit at 10 years or older. The method of a self-administered questionnaire was selected in order to ensure that all information collected was as genuine as possible while protecting the privacy of participants.

We explained the purpose and content of the research to all participants before undertaking the survey. The mean time for completing questionnaires was 40 min. To minimize data errors and missing data, each questionnaire was double-checked by two trained quality control personnel on-site once completed. Quality control personnel were either undergraduate students recruited from Yunnan Western University, a local college in Lincang, or graduates majoring in Psychiatry or Public Health from Kunming Medical University. A total of 55 quality control staff received centralized training on November 30, 2019. Specific information such as study design, calculation of sample size, and participant information has been previously published (*Ran et al., 2020*).

## Participant information

The questionnaire captured general personal information about the individual such as their gender, grade, type of school attended, ethnicity, level of attendance, social support, parental marital status, parental labor migration status, and family income bracket.

## Modified adolescents self-harm survey (MASHS)

The MASHS was developed by *Feng (2008)* to measure the method and severity of self-harming behavior in an adolescents' life time. It included 18 method of self-harm behaviors and 1 open-ended question. The frequency of each self-harm behavior is categorised into four levels (0, 1, 2–4, 5 and more). The seriousness of injuries sustained has been separated into five levels (none, mild, moderate, severe, and extremely severe), which is represented in numeric values from 0 to 4 in ascending order respectively, "none" meaning no damage to the body and "extremely severe" meaning that hospitalization was required as a result of the injuries. In the current study, repeated NSSI was defined as the frequency of self-harming behavior was 2–4 or 5 and above. The severity of NSSI was divided into two groups: The patients with the scores of self-harming degree between 1–2 points were assigned to mild or moderate group, while those with the scores of self-harming degree between 3–4 points were assigned to severe group. The major aim of our current study was considering the NSSI as a behavior rather than a diagnosis. Meanwhile, we emphasized that the significant value of intervention method taken in early stage for the adolescents with NSSI. Therefore, in our current study, we defined the NSSI as the participants had self-harming behavior at least once in order to identify the high risk
patients in the population. Previous studies have reported to evaluate the degree of self-harming by using the product of frequency and degree of physical in MASHS, however, we prefer to consider the frequency and degree of physical injury as two independent risk for evaluation so that we could find whether different factors had influence on these two dimension.

### Mobile phone addiction index (MPAI)

The mobile phone addiction index (MPAI) was revised by *Leung (2008)*. It was translated into Chinese edition by *Huang et al. (2014)* to diagnose cell phone addiction in adolescents and college students. It comprises 17 items, each scored on a scale of 1 to 5, with a higher score indicating a higher degree of mobile phone addiction. The full scale can be divided into four subscales: the inability to control cravings subscale (ICCS), the feeling anxious and lost subscale (FALS), the withdrawal and escape subscale (WES), and the productivity loss subscale (PLS). The ICCS assesses the individual's capability to control the amount of time they spend on their mobile phone. The FALS assesses the individual's ability to adapt to the adverse emotional reactions towards not having a normal level of access to their mobile phone. The WES assesses the individual's use of their mobile phone to immerse themselves in the cyber world and escape reality. The PLS assesses the degree to which their ability to work or study is decreased due to their excessive mobile phone use.

In addition, the item 3–6, and item 8, 9, 14, 15 are also MPA screening questions. So we divided MPA into two groups: (1) none (a cumulative score over these eight questions of three points or less); (2) yes (a cumulative score over these eight questions of more than three points).

### Statistical analysis

Data were checked using Epidata 3.0 for double entry and consistency. Statistical analyses were conducted using R software (version 3.3.3). Participant demographic characteristics were explored using descriptive analysis. An unequal probability sampling weight adjustment method was used to account for the possibility of interrelationships among the participants introduced by cluster sampling. To adjust for the effect of clustering, the 'survey' and 'lavaan.survey' R packages were used.

A single factor logistic regression model was used to explore the factors associated with NSSI, repeated NSSI, and severe NSSI. The univariate logistic regression analysis utilized a lower threshold of $p \le 0.1$ to avoid omitting important factors. Associations between scores obtained in the MPAI and its four subscales and NSSI, repeated NSSI, and severe NSSI were measured using a multivariate binary logistic regression model. Two-tailed significance was considered at a level of $p < 0.05$.

## RESULTS

### General participant characteristics

A total of 3,150 participants completed the survey. Of whom, four were excluded because their age was less than or equal to 10 years and 427 were excluded due to unclear or missing information on parental educational level or marital status. The final sample size

**Table 1 General sample characteristics (*n* = 2,719 students in Lincang Yunan).**

| Variable | *n* (%) | Mean (standard deviation) |
|---|---|---|
| Age | | 13.42 (2.17) |
| Gender | | |
| Boys | 1,246 (45.83) | |
| Girls | 1,473 (54.17) | |
| Ethnicity | | |
| Han | 1,816 (66.79) | |
| Other ethnic groups | 280 (10.30) | |
| Yi | 320 (11.77) | |
| Dai | 185 (6.80) | |
| Lahu | 118 (4.34) | |
| Grade | | |
| Primary school | 918 (33.76) | |
| Middle school | 939 (34.53) | |
| High school | 862 (31.71) | |
| Residency | | |
| Urban | 1,268 (46.63) | |
| Rural | 1,451 (53.37) | |
| Level of attendance | | |
| Day school | 1,059 (38.95) | |
| Boarding school | 1,660 (61.05) | |
| Parental marital status | | |
| Married | 2,456 (90.33) | |
| Divorced/Widowed | 163 (5.99) | |
| Re-married | 100 (3.68) | |
| Mobile phone addiction | | |
| No | 2,417 (88.89) | |
| Yes | 302 (11.11) | |
| MPAI | | |
| Inability to control cravings | | 16.79 (7.36) |
| Feeling anxious and lost | | 6.82 (3.92) |
| Withdrawal and escape | | 7.22 (3.74) |
| Productivity loss | | 5.36 (2.79) |
| NSSI | | |
| No | 1,438 (52.89) | |
| Yes | 1,281 (47.11) | |
| Repeated NSSI | | |
| No | 445 (34.74) | |
| Yes | 836 (65.26) | |
| NSSI severity | | |
| Mild or moderate | 812 (63.39) | |
| Severe | 469 (36.61) | |

Note:
NSSI, Non-suicidal self-injury; MPAI, Mobile phone addiction index; Repeated NSSI was defined as the frequency of self-harming behavior was 2–4 or 5 and above. NSSI severity: mild or moderate (1–2 points) and severe (3–4 points).

Table 2 Univariate and multivariable logistic regression model for NSSI.

| Covariate | Univariate | Model 1 | Model 2 |
|---|---|---|---|
| | OR (90% CI) | OR (95% CI) | OR (95% CI) |
| age (+1) | 1.275 [1.198–1.357]* | 1.136 [1.014–1.271]** | 1.110 [0.982–1.254] |
| Gender (Ref: Boys) | 1.347 [1.232–1.474]* | 1.251 [1.147–1.365]** | 1.228 [1.132–1.332]** |
| Residency (Ref: Town) | 1.716 [1.393–2.114]* | 1.040 [0.798– 1.354] | 1.024 [0.790–1.328] |
| Grade (Ref: Primary school) | | | |
| Middle school | 2.479 [1.488–4.130]* | 1.867 [0.680–5.128] | 1.396 [0.555–3.516] |
| High school | 3.662 [2.560–5.237]* | 2.005 [0.733–5.486] | 1.228 [0.491–3.069] |
| Ethnicity (Ref: other) | 1.017 [0.817–1.266] | | |
| Parental marital status (Ref: married) | | | |
| Divorced/Widowed | 1.196 [1.003–1.426]* | 1.324 [1.042–1.683]** | 1.238 [1.017–1.507]** |
| Re-married | 1.528 [1.228–1.901]* | 1.521 [1.116–2.073]** | 1.375 [0.967–1.954] |
| Father's education level (Ref: Primary school) | | | |
| Middle school | 0.892 [0.711–1.119] | 1.007 [0.771–1.314] | 1.051 [0.825–1.339] |
| High school | 0.693 [0.469–1.024] | 0.972 [0.695–1.360] | 1.071 [0.791–1.452] |
| College and above | 0.639 [0.482–0.847] | 0.863 [0.594–1.254] | 0.931 [0.651–1.330] |
| Mother's education level (Ref: Primary school ) | | | |
| Middle school | 0.807 [0.709–0.919] | 0.970 [0.878– 1.073] | 0.979 [0.855–1.121] |
| High school | 0.624 [0.436–0.892] | 0.838 [0.630–1.116] | 0.893 [0.658–1.210] |
| College and above | 0.623 [0.451–0.860] | 0.968 [0.763–1.228] | 1.089 [0.860–1.379] |
| Level of attendance (Ref: Day school) | 2.436 [1.744–3.401]* | 0.785 [0.562–1.096] | 0.757 [0.570–1.006] |
| Mobile phone addiction | 5.471 [4.382–6.832]* | 4.280 [3.480–5.266]** | |
| Inability to control cravings (+1 point) | | | 1.071 [1.060–1.083]** |
| Feeling anxious and lost (+1 point) | | | 1.033 [0.997–1.071] |
| Withdrawal and escape (+1 point) | | | 1.055 [1.040–1.070]** |
| Productivity loss (+1 point) | | | 1.071 [1.033–1.111]** |

Notes:
* $P \leq 0.1$.
** $P < 0.05$.
Model 1: Multivariate logistic regression analyses on associations of mobile phone addiction with NSSI, controlling for age, gender, residency, grade, parental marital status, parental education level, level of attendance, mobile phone addiction.
Model 2: Multivariate logistic regression analyses on associations of the four specific dimensions of mobile phone addiction with NSSI, controlling for age, gender, residency, grade, parental marital status, parental education level, level of attendance, each of the four MPAI subscales.

was 2,719, making the response rate 86.3%. Table 1 shows the characteristics of the participants. The average age was 13.42 ± 2.17 years. A total of 1,281 participants reported NSSI experience, making the prevalence 47.11% (95% CI [36.2–58.0%]), of whom 65.26% (95% CI [58.2–72.0%]) had repeated NSSI and 469 (36.61%) had severe NSSI. Further, 302 students, accounting for 11.11% (95% CI [6.7–18.0%]) of the sample, were found to have MPA. The average score of each MPAI subscale varied from the PLS (5.36 ± 2.79) to the ICCS (16.79 ± 7.36).

## Prevalence and associations of NSSI

Table 2 lists the results of the univariate and multivariate logistic regression of NSSI. All factors aside from ethnicity and parents education level (less than college) were found

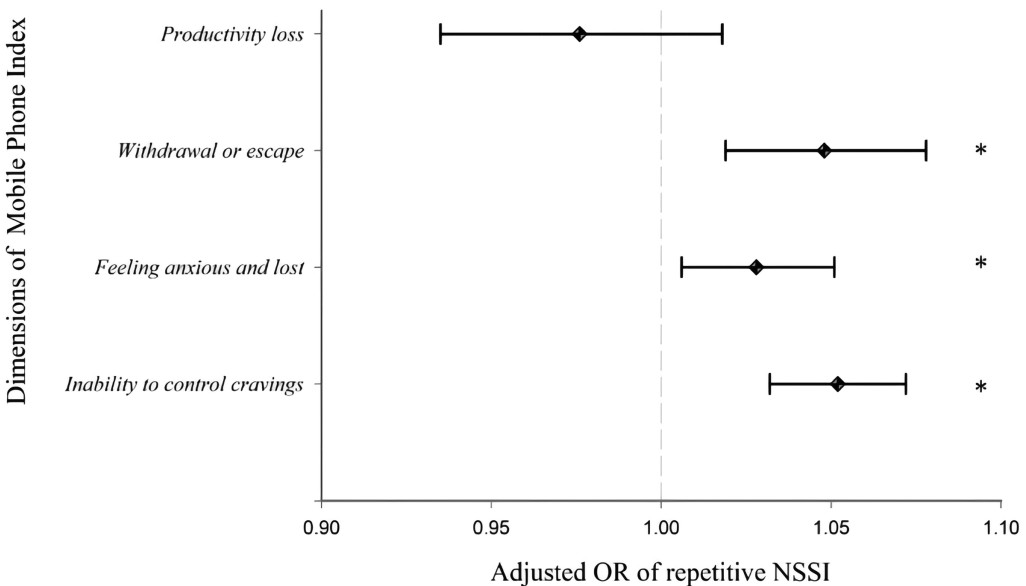

Figure 1 **Adjusted OR of repeated NSSI.** Multivariate logistic regression model for repeated NSSI. Controlling for age, gender, level of attendance, each of the four MPAI dimensions.

to be potentially associated with NSSI. After multivariate adjustment, the associations between NSSI and age, gender, parental marital status, and MPA were still statistically significant, as evidenced by a significance of $p < 0.05$. Participants who were found to have a MPA were 4.280 times (95% CI [3.480–5.266]) more likely to also have NSSI. A multivariate logistic regression model was further used to include the four MPAI subscales. These results suggest that the four MPAI subscales, except for the FALS, were associated with NSSI.

### Univariate and multivariate analysis of repeated NSSI

We analyzed the association between the MPAI score and the score obtained on each of the four MPAI subscales and both repeated NSSI and severe NSSI. Taking repeated NSSI as the dependent variable and other possible covariates into consideration, Fig. 1 demonstrates that the ICCS, the FALS, and the WES were all statistically significantly and positively correlated with repeated NSSI. In other words, for every one point increase in subscale score, the risk of repeated NSSI increases by 1.052. (inability to control cravings; 95% CI [1.032–1.072]), 1.028 (feeling anxious and lost; 95% CI [1.006–1.051]), and 1.048 (withdrawal and escape; 95% CI [1.019–1.078]).

### Univariate and multivariate analysis of the severity of NSSI

Figure 2 shows that with each increasing point on the same three subscales, the risk rate of severe NSSI increases by 1.048 (inability to control cravings; 95% CI [1.029–1.068]), 1.033

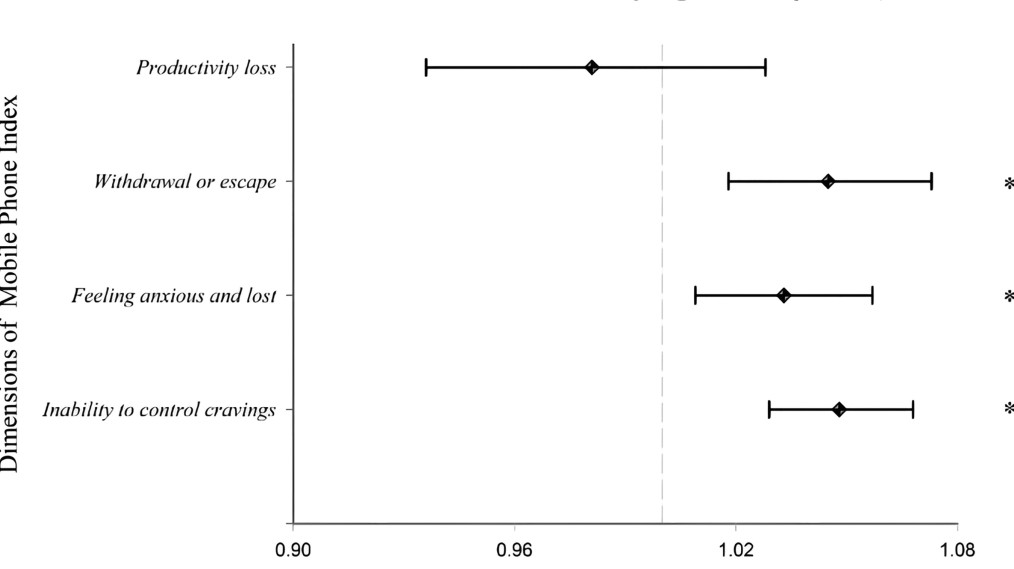

**Figure 2 Adjusted OR of severe NSSI.** Multivariate logistic regression model for NSSI severity. Controlling for gender, parental marital status, parental education level, level of attendance, each of the four MPAI dimensions.

(feeling anxious and lost; 95% CI [1.009–1.057]), 1.045 (withdrawal and escape; 95% CI [1.018–1.073]).

## DISCUSSION

The results of this study illustrated that the lifetime prevalence of NSSI was 47.11% (95% CI [36.2–58.0%]), which was similar to the results of a study conducted in the USA, and the findings of Feng et al. (*Burke et al., 2015*; *Feng, 2008*) using the same index, but higher than the results of a domestic meta-analysis (16.6%) and some other international studies (7–13%) (*Portzky, Wilde & Heeringen, 2008*). These differences are likely to be due to the use of different scales and evaluation criteria. The high prevalence in our study is due to the NSSI definition: the presence of one or more episodes of self-harm behaviour in any way. Although the recommended diagnostic criterion in the DSM-5 is at least five episodes of NSSI in the past year. However, NSSI has not yet been formally included in the DSM as a separate psychiatric disorder, so this study used lower inclusion criteria to increase sensitivity to self-harm in children and adolescents so that prevention and intervention could be provided earlier.

Nevertheless, these findings reflect the high prevalence of NSSI among adolescents. Our study has revealed that 65.26% of the adolescents with a history of NSSI had repeated NSSI, and 36.61% had severe NSSI. Those with severe NSSI are more likely to be re-admitted to hospital due to self-harm and attempted suicide, or even suicide death, in the future. These two high-risk factors also were applied to psychological risk assessment. This may suggest that individuals with a history of self-harm are more likely to exhibit other psychiatric disorders such as anxiety, emotional disorder, and borderline personality

disorder, needing further diagnosis and treatment (*Gardner et al., 2019*; *Knipe et al., 2019*). Therefore, further research is needed to understand the mechanisms of NSSI to better prevent and control it.

This study also found that the prevalence of NSSI increases with age, with participants' age ranged from prepubescent to adolescent (*i.e.*, students aged 10 to 18). *Steinberg et al. (2008)* reported that "impulsive choices" are most common in early- to mid-puberty, which may influene the prevalence of NSSI in puberty. Also, girls were found to be more susceptible to NSSI than boys, which is consistent with the findings of multiple meta-analyses conducted by *Lang & Yao (2018)* and *Bresin & Schoenleber (2015)*. The reasons for this gender difference have not yet been fully confirmed. Girls may be more likely to experience frustration and depression than boys, and need emotional management. NSSI, as an emotion regulation strategy, is often used to alleviate negative emotions (*Albores-Gallo et al., 2014*). As a result, different intervention tactics may be employed more frequently by people of different genders. NSSI was also suggested to be associated with parental marital status. Compared with children from stable families, those with divorced, widowed, or remarried parents are more likely to have a history of NSSI, which is consistent with previous findings (*Hawton, Saunders & O'Connor, 2012*; *Lan et al., 2019*). The parental relationship is an indispensable environmental factor in adolescent development as well as an underlying cause of many behavioral problems. A dysfunctional family environment could play a crucial role in adolescent NSSI through impacting self-esteem, impulse control, emotional regulation, interpersonal communication, and creating the need for coping mechanisms (*Cassels et al., 2018*; *Kelly, 2000*). Improving family relations may serve to reduce NSSI in later years, so it may be crucial to focus on the family environment and intervention to improve it in order to reduce NSSI in adolescents.

The popularity and customizability of mobile phones has made them a necessity for young people, facilitating everyday social interaction and providing a source of entertainment. Modern smartphones allow apps to be downloaded in order to use them for more work and study related tasks such as online courses and information exchange with schools, which in turn raises the perception of the necessity of their use in modern society.

Many international academics have identified this dependence as an addiction (*Beard & Wolf, 2001*; *Billieux, Linden & Ceschi, 2007*), which is manifested as the excessive use of mobile phones, resulting in an adverse effect on work, study, and life, as well as a series of physical and psychological reactions signifying discomfort and withdrawal if mobile phone use is prohibited. As such, the misuse of mobile phones by young people and the associated problems of MPA must be explored. This study aims to better understand the relationship between repeated NSSI and severe NSSI by investigating the influence of the four MPAI subscales and offering preventive ways to guide students in appropriate mobile phone use. The prevalence of MPA in this study was 11.11%, which is lower than that reported by *Ge & Zhu (2014)*. This may be due to the study location being a poverty-stricken area where mobile phone ownership among young people is lower than that of more developed cities. We also included students living on campus, where mobile phone use is prohibited in school. This may also lower the rates of phone use and phone dependence.

There is still no universally accepted standard scale for assessing or diagnosing MPA, and the results vary widely from study to study. More uniform and large-scale studies are needed in this area. In our study, MPA was strongly associated with the prevalence of NSSI. MPA, and other addictions such as alcohol, tobacco and substance abuse, have been positively associated with NSSI. People who are dependent on mobile phone use show withdrawal symptoms when denied access to their phones, leading to anxiety, depression and negative emotions. In addition, MPA itself can have a negative impact on mood.

The stress from interpersonal problems can lead to depression (*Oshima et al., 2012*), thereby increasing the risk of NSSI. Meanwhile, people with depression may use mobile phones as an escape mechanism to relieve negative emotions and symptoms by obtaining a sense of security (*Park et al., 2019*). The interaction between MPA and depression as well as each factor independently can increase the risk of NSSI. Individuals experiencing issues with interpersonal social interaction, for instance, social isolation or a lack of social support, are more inclined to use mobile phone games, videos, and other applications to kill time, escape reality, and seek recognition or validation on social media to combat loneliness. The motives for self-harm also include escape from unbearable social situations and using NSSI as a means of showing others their degree of suffering to gain attention or recognition (*Hawton, Saunders & O'Connor, 2012*). As a result, MPA and NSSI may share similar motivations, resulting in a significant correlation between the two. The coincidence of repeated and severe NSSI can signify the presence of more serious mental health problems (*Hawton et al., 2020*). This, in turn, helps to explain the significant association between severe NSSI and high scores on the WES and the FALS. High impulsivity has also been positively correlated with MPA in previous studies, reflecting that individuals with high impulsivity are less likely to control the urges to use their mobile phones, resulting in a higher score on the ICCS (*Mitchell & Hussain, 2018*). Our findings suggest that a high score on the ICCS is significantly associated with repeated and severe NSSI. This may also be related to poor levels of impulse control in adolescents. Studying impulsivity's effect was outside the scope of the current investigation, but it will be investigated in future research.

Our primary finding is that MPA is not only associated with NSSI, but is predominantly manifested on two subscales of the MPAI, namely "ICCS" and "WES". Both subscales were significantly associated with the occurrence of NSSI, repetitive NSSI and severe NSSI. If MPA and NSSI are underpinned by the same factors, it would be useful to examine the role of mediating and interacting factors associated with these two behaviours. Children and adolescents aged 10 to 18 years were recruited for this study, highlighting that these findings should alert the government and schools to pay closer attention to potential problems associated with inappropriate mobile phone use, such as NSSI, which may have a greater impact on younger pupils. There may be a need to establish a monitoring system for NSSI, with regular monitoring, screening and intervention when needed. In establishing intervention methods, more targeted measures can be developed based on different factors affecting different populations, such as improving impulse control in adolescents, making adjustments for depression and anxiety, or understanding the causes and consequences of adopting an avoidant approach to social stress. In addition,

interventions should emphasise the importance of developing coping strategies that are more positive than self-harm.

## CONCLUSIONS

This cross-sectional study demonstrates that the prevalence of NSSI is relatively high. MPA is an important risk factor, specifically, the MPAI subscales for the inability to control cravings, withdrawal and escape, and feeling anxious and lost were all strongly associated with repeated and severe NSSI. These findings indicate that monitoring and intervention for MPA may partially reduce the prevalence of NSSI, repeated NSSI, and severe NSSI.

**Limitations:** The survey was conducted in Lincang, with cross-sectional recruitment and data collected using a self-reported questionnaire. In the sample, there could be an influence of personal memory bias. As a result, significant relationships between behaviors and survey subscales must be substantiated before being accepted. To investigate the causal links between these parameters, more longitudinal follow-up is required. Because no formal measurement of mobile phone usage duration or frequency was included in the study, mobile phone use was judged subjectively, resulting in an unreliable assessment of MPA. Further longitudinal research is required to confirm the association between mobile phone use frequency, total duration of use, and MPA. The time period defined in the two scales for assessing mobile phone dependence and non-suicidal self-injury in our study is the situation of the entire life cycle of the subject, and the relationship between the occurrence of self-injury and the chronological order of occurrence of mobile phone dependence has not been specifically evaluated, which requires more detailed research in the future to further prove the views of this study. Finally, the motivations behind MPA and NSSI were discussed, and the possible commonality between these mechanisms has not yet been addressed.

### Funding

This work was supported by the Yunnan Health Training Projects of Highly Level Talents (D-2017048), the Scientific Research Fund Project of Yunnan Provincial Department of Education (2018JS198), the Yunnan Applied Basic Research Projects-Kunming Medical University Union Foundation (2018FE001(-132)), the Top Young Talents of Yunnan Ten Thousand Talents Plan (YNWR-QNBJ-2018-286). The funders had no role in study design, data collection and analysis, decision to publish, or preparation of the manuscript.

### Grant Disclosures

The following grant information was disclosed by the authors:
Yunnan Health Training Projects of Highly Level Talents: D-2017048.
Scientific Research Fund Project of Yunnan Provincial Department of Education: 2018JS198.
Yunnan Applied Basic Research Projects-Kunming Medical University Union Foundation:

 

2018FE001(-132).
Top Young Talents of Yunnan Ten Thousand Talents Plan: YNWR-QNBJ-2018-286.

## Competing Interests

The authors declare that they have no competing interests.

## Author Contributions

- Rui Wang conceived and designed the experiments, performed the experiments, analyzed the data, prepared figures and/or tables, authored or reviewed drafts of the article, and approved the final draft.
- Runxu Yang conceived and designed the experiments, performed the experiments, analyzed the data, prepared figures and/or tables, authored or reviewed drafts of the article, and approved the final draft.
- Hailiang Ran performed the experiments, analyzed the data, prepared figures and/or tables, and approved the final draft.
- Xiufeng Xu conceived and designed the experiments, performed the experiments, prepared figures and/or tables, authored or reviewed drafts of the article, and approved the final draft.
- Guangya Yang performed the experiments, prepared figures and/or tables, and approved the final draft.
- TianLan Wang performed the experiments, prepared figures and/or tables, and approved the final draft.
- Yusan Che performed the experiments, prepared figures and/or tables, and approved the final draft.
- Die Fang performed the experiments, prepared figures and/or tables, and approved the final draft.
- Jin Lu conceived and designed the experiments, performed the experiments, analyzed the data, prepared figures and/or tables, authored or reviewed drafts of the article, and approved the final draft.
- Yuanyuan Xiao conceived and designed the experiments, performed the experiments, analyzed the data, prepared figures and/or tables, authored or reviewed drafts of the article, and approved the final draft.

## Ethics

The following information was supplied relating to ethical approvals (*i.e.*, approving body and any reference numbers):

The Third People's Hospital of Lincang Ethics Committee approved this research (2019-01).

## Data Availability

The raw measurements are available as a Supplemental File.

## Supplemental Information

Supplemental information for this article can be found online at http://dx.doi.org/10.7717/peerj.14057#supplemental-information.

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
