# Peer review of "Mobile phone addiction and non-suicidal self-injury among adolescents in China"

_PeerJ, doi:10.7717/peerj.14057_

## Round 0.1 · original submission · Major Revisions

Thank you for submitting the manuscript to PeerJ. It has been reviewed by experts in the field and we request that you make major revisions before it is processed further.

We look forward to hearing from you soon.

Best wishes,

Badicu Georgian, Ph.D

·

Basic reporting

English needs to be revised by a fluent English speaker. Literature references should be updated

Experimental design

This is an original and relevant study. Research question is well defined and methods are described in great detail

Validity of the findings

The study revealed impact and novelty. Needs some improve in language before acceptance

Reviewer 2 ·

Basic reporting

I found that the topic of this article was adequately introduced, except for the following issues:

1. What do you mean when stating "There are two main mechanisms for NSSI: aggression and violent outbursts and coping mechanisms for depression or withdrawal" (lines 60-62)? If you want to explain the mechanisms of NSSI, I think you should add more information, even if synthetically, about the NSSI functions (eg., the four function model by Nock and Prinstein and other theories).

2. No references are provided about the international prevalence of NSSI, but only data from China and Taiwan. Moreover, I think that the authors may specify the differences in NSSI rates between clinical and community samples. Thus, I think more references are needed about these aspects.

3. Your introduction to mobile phone addiction needs more detail. Please provide more recent references about this topic and base its definition on literature.

Figures and table need to be improved:

Table 1: I think it would be clearer if you specified the criteria for “Repeated NSSI” and “NSSI severity”

Table 2: the title and the legend are not clear (it should be clarified which are the predictors and the outcome of the regression model); the p-values are missing and should be added. It is written "Molde 1" and "Molde 2" instead of "Model".

Figures 1 and 2: the p-values are missing and should be added.

Experimental design

Concerning the research question, it is stated how the research fills a knowledge gap, but I think it is necessary to explain what the Mobile Phone Addiction Index is and what it measures when you name it for the first time in line 94, otherwise, your second research hypothesis is not clear.

I think that several methodological information is missing:
1. I suggest that you improve the description of the recruitment phase by answering, for example, important questions like: Where is Lincang? Which students were recruited? What age? From which schools? What were the inclusion and exclusion criteria? How was the sample size established? Without this information, the study is very unlikely to be replicable by other researchers.

2. We know from what you state in lines 110-111 that the severity of NSSI measured by the MASHS depends on the number of NSSI episodes, but we have no information about the time of these episodes: are you talking about current or past NSSI or both? Please, specify. Furthermore, I think you should explain more clearly if the scoring system depends on both the “severity” and the “degree of physical injury” or not. In this way, the reader may understand more clearly what “Repeated” self-harm refers to. Finally, I understand that you consider a history of NSSI when at least one episode of NSSI occurred; but I think you should discuss your choice in the context of the DSM-5 proposed diagnostic criteria of NSSI (at least five episodes of NSSI in the last year)

3. Concerning the Mobile Phone Addiction Index, are there cut-off scores for the full scale and the subscales?

Validity of the findings

The topic of this study is of interest to the field. Unfortunately, its replication is difficult due to the lack of necessary methodological information.

The statistical robustness of the study is weak for the following reasons:
1. I think it is important to consider and discuss that the statistical significance for the univariate logistic model was set at p </= 0.1, which provides a weak level of evidence.
2. It is stated that 11.11% of the sample was found to have a Mobile Phone Addiction (line 150) but it is not explained on what basis this percentage is estimated.
3. The prevalence of NSSI is estimated based on at least one episode of self-harm, which is insufficient to talk about NSSI according to the DSM-5 criteria
4. The p-value is never reported in any regression model. In this way, the statistical significance of the results is not demonstrated.

The validity of the conclusions is not fully evaluable due to the uncertain reliability of the results (as written above)

Additional comments

No additional comments

Reviewer 3 ·

Basic reporting

This paper deals with the interesting and important problem of mobile phone addiction in young people. Its consequences are various, among others it may be related to the Non-Suicidal Self-Injury. I think that results are imortant but I have some doubts, reservations and questions.

Experimental design

First of all, there is no information on how the recruitment for the study was conducted. Were they volunteers, was it a randomized study, and if so, how large was an output group? Whether it was determined how large the group should be in order for the results to be reliable
What was the percentage of respondents in relation to all teenagers in the region?
In Line 146 Authors wrote: "The final sample size was 2719, making the recovery rate 86.3%. In my opinion this is not a recovery rate, but the number of participants who met the criteria of inclusion.

Validity of the findings

The applied statistical methods allowed to assess the general trend of the relationship between NSSI and mobile phone addiction. The adopted significance level p <0.1 indicates only a statistical tendency. It is not possible to draw conclusions about significance on their basis, such results indicate certain relationship that in other conditions, e.g. a larger sample, could be statistically significant. Therefore, the statement “After multivariate adjustment, the associations between NSSI and age, gender, parental marital status, and mobile phone addiction were still statistically significant” is not fully substantiated. The most commonly used significance level is p≤0.05.
The age range of the respondents is very wide (10-18 years), between adolescents in puberty time and after it, etc. The statement that the influence of age as a confounding factor was taken into account seems insufficient, as it would be worth analyzing in more depth. Maybe the boundary of described behaviors could be found?
These relationship are discussed partially by the Authors in the Discussion, citing the results of other studies, but they do not compare with their own results.
Did 10-year-old children fill in the questionnaires themselves or with the help of their guardians. This may affect the results and should be explained
In my opinion, the information provided by the Authors in the "Discussion" (lines 231-233) should be included in the description of the research group (Materials and Methods).

Additional comments

Minor remarks
line 161 In "Methods" Authors wrote that there were 4 subscales. According to the results, two subscales were not correlated with NSSI and the other two were correlated. Thus, the sentence "These results suggest that all subscales except for the feeling anxious and lost subscale were associated with NSSI" is badly worded
Line 174 - % and percent means the same. 63.4% percent is incorrect. It should be 63.4% or 63.4 percent
Line 226 The authors used a mental abbreviation: “Impact of four MPAI subscales...” This is not an impact of subscales, rather relationship between various aspects of cellphone addiction and repeated NSSI and severe NSSI
In Fig 1 and 2 caption should be indicated for what factors OR was adjusted
“OR adjusted for......”It is not enough to explain it in the text
Table 2 typing error (molde 1 instead of model 1)

---

## Round 0.2 · Minor Revisions

Thank you for submitting the manuscript to PeerJ. It has been reviewed by experts in the field and we request that you make minor revisions before it is processed further.

We look forward to hearing from you soon.

Best wishes,

Badicu Georgian, Ph.D

Reviewer 2 ·

Basic reporting

The article has been improved, both in the content and in the formal aspects. The literature has been broadened.I think there are some points that should be further improved before publication.

Why do the authors cite the author Taliaferro but report the reference by Muehlenkamp (lines 58-59)? Which one is correct?

In lines 83 and 84, the same phrase is repeated twice. Please, delete one of the two.

Experimental design

The two hypotheses of the work seem to be quite redundant because very similar, with the second included in the first. The authors can consider simplifying the description of the aim of the work by using the first hypothesis and mentioning the mobile phone addiction scale with its 4 subdomains.

The authors should pay attention to the double mentioning of the Ethical approval of the study by two different ethics committees (line 123 and line 135).

What does “Zhikong” (line 144) mean? The sentence is unclear. Please, correct.

I think that the paragraph about the MASHS (lines 154-165), although improved, is still quite unclear.
- When the authors report “The MASHS was developed.. to measure the way and severity of self-harming” (line 155), what do they mean by “the WAY”? Do they refer to the method of self-injury? Please, clarify.
- The authors have changed the “severity of the self-harm” (line 110 in the previous version) with “the times of self-harm” (line 157), which, in my opinion, is better to define as the “FREQUENCY” of self-harm behavior.
- In my opinion, it is not fully comprehensible what the scoring system 0-4 (line 159) refers to and how it is calculated. It is plausible that it refers to the “degree of physical injury” because classified over 5 levels and, therefore, on a 0-4 scale (0=none; 1=mild; 2=moderate; 3=severe; 4=extremely severe), while the frequency is expected to be scored on a 0-3 scale because classified over 4 levels (0=0 times; 1=1 time; 2=2-4 times; 3=5 times and above). But the severity score could also be cumulative for both the frequency and the degree of physical injury. I think that this information needs to be explained in an explicit and non-confusing way. Clear definitions of what you mean by REPEATED NSSI and SEVERE NSSI should be given because they are very central to the aim and the results of the work.

Validity of the findings

I think it could be useful to mention in the Limitations section the absence of data about the historical time of the episodes of self-injury, because in this way, hypothetically, NSSI could have happened in very different periods of time than mobile phone overuse.

Reviewer 3 ·

Basic reporting

After the corrections made, the manuscript is more valuable

Experimental design

Improved according to the reviewer's suggestions

Validity of the findings

Manuscript is improved according to the reviewer's suggestions

Additional comments

Improved according to the reviewer's suggestions

---

## Round 0.3 · accepted · Accept

I am writing to inform you that your manuscript has been Accepted for publication. Congratulations!